# DEEP CONCEPT REMOVAL

## ABSTRACT

We address the problem of concept removal in deep neural networks, aiming to learn representations that do not encode certain specified concepts (e.g., gender etc.) We propose a novel method based on adversarial linear classifiers trained on a concept dataset, which helps remove the targeted attribute while maintaining model performance. Our approach Deep Concept Removal incorporates adversarial probing classifiers at various layers of the network, effectively addressing concept entanglement and improving out-of-distribution generalization. We also introduce an implicit gradient-based technique to tackle the challenges associated with adversarial training using linear classifiers. We evaluate the ability to remove a concept on a set of popular distributionally robust optimization (DRO) benchmarks with spurious correlations, as well as out-of-distribution (OOD) generalization tasks.

## 1   INTRODUCTION

It is well known that deep neural networks encode the information of various concepts in the latent representation space Bau et al. (2017). The ability to remove a specified concept (e.g., by the model trainer or user) from the learned representation and the model is crucial in many ways. For example, some concepts may represent detrimental features, such as ones that are not relevant to the downstream task, but are nevertheless spuriously correlated with the target variable, e.g., the background for classifying the type of animal Beery et al. (2018); some of the attributes might represent information that is once informative but nonetheless is no longer so; others may represent sensitive features, such as gender or race, which are undesirable for the model to correlate with. Removing these features will produce more robust, generalizable, and fair models that are oblivious to the presence of them. In this paper, we consider the problem of *concept removal* — we want to learn deep neural representations that do not encode a certain specified attribute.

A large set of existing literature is concerned with adversarial concept removal Elazar & Goldberg (2018); Ye et al. (2021); Moyer et al. (2018). These methods seek to learn representations that are statistically independent of sensitive attributes, thus preventing any potential inference, with an adversarial classifier serving as a proxy to measure the mutual information. While these methods help to mitigate the bias, they may be limited in generalizing to out-of-distribution (OOD) data, as the independence is subject to the specific distribution of the input data. Furthermore, these methods require labeled sensitive attributes in the training dataset.

In this paper, we propose an adversarial method that relies on a *concept dataset* instead. We borrow this notion from the interpretability literature Kim et al. (2018); Chen et al. (2020b); Crabbé & van der Schaar (2022), where a concept dataset refers to a set of examples that are chosen or curated to represent a specific concept of interest. For instance, to determine whether the classification of a "zebra" relies on the concept "stripes" they collect a set of images with striped patterns. These images are used to construct a linear concept classifier by separating them from some random images in the latent space of a pretrained neural network. A concept dataset is also potentially cheaper to obtain since it can be composed of publicly available or synthetic data. While interpretability methods are primarily concerned with detecting whether a classifier relies on a certain concept, our goal is to mitigate this effect through adversarial training.

Departing from the literature where adversarial classifiers are typically applied to the output of neural network's penultimate layer (see also Madras et al. (2018); Ganin & Lempitsky (2015)), our paper proposes a method *Deep Concept Removal* that simultaneously targets representations of

deeper layers along with the penultimate one. In Section 4, we conduct extensive experiments and confirm that our method works best when adversarial classifiers are applied to the widest layers in the network that precede contraction. In Section 5, we demonstrate how it improves robustness to distributional shifts and helps to generalize to out-of-distribution data.

## 2 Formulation and Preliminaries

Suppose, we have a downstream task, where we want to predict a label $Y \in \mathcal{Y}$ from the input $X \in \mathcal{X}$. We are given a neural network model in the form $f_k(h_k(X; W); \theta)$, where $W$ are the parameters of the hidden layers, $h_k(X; W)$ is the output of the $k$th layer, which we treat as representation, and $f_k(\cdot; \theta)$ is a classifier on top of the extracted representation, parameterized by $\theta$. Without carrying the index $k$, suppose the prediction has the form $\hat{\Pr}(Y \mid X) = f(h(X))$. Apart from the main training dataset $(X, Y)_{i=1}^N$, we are given a concept dataset $(X_i^C, Y_i^C)_{i=1}^{N_C}$, where $Y_i^C = 1$ when the instance is from the concept class and $Y_i^C = 0$ otherwise.

### 2.1 Concept Activation Vectors

Kim et al. (2018) propose a method that tests if a concept is used by an ML classifier, that goes beyond correlation analysis. The flexibility of their method allows users to create a concept (not necessarily used during model training) by providing concept examples and linearly separating them from random instances which together constitute the *concept dataset* $(X_i^C, Y_i^C)_{i=1}^{N_C}$. The Concept Activation Vector (CAV) $v_C$ is estimated as a linear classifier on top of the hidden layer, i.e., a normal vector to a hyperplane separating the corresponding representations. Then, they use this concept vector to calculate directional derivatives to see if the output of the network is aligned with a concept. Intuitively speaking, we shift the representation of input $X$ in the direction of $v_C$ (making it more relevant to the concept) and see how the output changes. The first-order approximation yields the sensitivity score

$$S_C(X) = \lim_{\epsilon \to 0} \frac{f(h(X) + \epsilon v_C) - f(h(X))}{\epsilon} = \frac{\partial f(h(X))}{\partial h^\top} v_C \,. \tag{2.1}$$

Based on the sensitivity score, they introduce Testing with Concept Activation Vectors (TCAV). Focusing on the sign of the sensitivity score, they utilize statistical significance testing to determine how often the sensitivity is positive for instances from a specific class. This analysis allows us to quantify the degree to which a model relies on it when making decisions.

Kim et al. successfully apply TCAV in various contexts. As an illustrative example, they first construct the "stripes" concept by providing examples of striped patterns and random images for non-concept instances. TCAV shows that this concept is crucial for recognizing "zebra" class, unlike, say, the class "horse". Some concerning examples reveal the model's reliance on sensitive information, such as gender, for classifying unrelated classes like "apron". Another study finds that TCAV identifies gender as a factor in differentiating between nurses and doctors, posing potential discrimination risks against protected groups Google Keynote (2019). Additionally, Kim et al. record instances where recognizing a ping pong ball relies on race. In the context of responsible machine learning, it is essential not only to detect such problematic behavior but also to mitigate it.

### 2.2 Adversarial Concept Removal in Representation Learning

Following Elazar & Goldberg (2018), adversarial concept removal consists of training simultaneously a downstream task classifier and an adversarial concept classifier by alternating between two objectives,

$$\min_{g_{adv}} \frac{1}{N} \sum_{i=1}^N \ell(g_{adv}(h(X_i)), Y_i^C)$$

$$\min_{h,f} \frac{1}{N} \sum_{i=1}^N \ell(f(h(X_i)), Y) - \frac{1}{N} \sum_{i=1}^N \ell(g_{adv}(h(X_i)), Y_i^C) \tag{2.2}$$

In the first equation, we are fitting the classifier $g_{adv}$ to predict the *attributed* concept indicators $Y_i^C$ given a fixed representation $h$. In the second part, given a fixed probing classifier $g_{adv}$, we simultaneously minimize the loss of a downstream task and maximize the loss of the adversarial classifier. In the ideal situation where $h(X)$ and $Y^C$ are independent, we should have the loss of the adversarial classifier close to the *chance* level. In other words, negative loss of the adversarial classifier is a proxy for the mutual information between $h(X)$ and $Y^C$. We emphasize that this approach requires the concept indicators $Y^C \in \{0, 1\}$ to be attributed in the training dataset (or at least a subset of it) Elazar & Goldberg (2018); Madras et al. (2018). These methods are designed in order to *sensor* the concept within a particular distribution, which does not mean that the concept is not included entangled with other features. For example, there are no guarantees that we can learn features that are transferable between instances of concept and instances not belonging to concept.

Instead, rely on the notion of concept sensitivity in terms that Kim et al. (2018) propose. There, a linear classifier $g_{adv}(h) = \sigma(v^\top h)$ is trained on the concept dataset $(X_i^C, Y_i^C)_{i=1}^{N_C}$ instead. The choice of a linear classifier is mainly motivated by its application in the concept-based interpretation literature. It has also become widely accepted in the deep learning literature to consider concepts as directions in the latent space Louizos et al. (2015). Furthermore, a linear classifier also has more chances to generalize to the original training data $X$, when we train it on the concept data $X_i^C, Y_i^C$.

We note that there are many difficulties associated with training adversarial classifiers, such as vanishing and unstable gradients Goodfellow (2016); Arjovsky & Bottou (2017). Although we do not know if these problems can be caused particularly by our choice of the discriminator, to the best of our knowledge linear adversarial classifiers have not been considered in the literature before. We introduce a new method to train them by modifying the loss function[1] and employing *implicit differentiation technique* Rajeswaran et al. (2019); Borsos et al. (2020); Lorraine et al. (2020).

## 3 METHODOLOGY

Recall that Kim et al. (2018) does not specify how exactly the linear classifier is obtained. Let us consider CAV as a penalized logistic regression estimator (for simplicity, we assume that the concept dataset is **balanced**, so that we do not have to include the bias)

$$v_{C,k,\lambda}^*(W) = \arg\min_v \frac{1}{N_C} \sum_{i=1}^{N_C} \ell_{BCE}\left(\sigma(v^\top h_k(X_i^C; W)), Y_i^C\right) + \frac{\lambda}{2}\|v\|^2, \qquad (3.1)$$

where $\ell_{BCE}(p, y) = -y\log p - (1-y)\log(1-p)$ is the binary cross-entropy loss, and $\sigma(x) = 1/(1 + e^{-x})$ is the sigmoid function. Here, $v_{C,k,\lambda}^*(W)$ is viewed as a function of $W$, and although we do not have an explicit analytical form, we can differentiate implicitly, see Section A.

What can be a good measure of the concept information encoded in the representation? If we use the loss function in (3.1), it goes back to the traditional adversarial approach. The effect of using the gradients of $v_{C,\lambda,k}^*$ vanishes thanks to the envelope theorem, and the optimization problem reduces to standard alternating procedure (for sake of completeness, we show this in detail in Section D).

Instead, we look at the parameter $v = v_{C,\lambda,k}^*(W)$ from the perspective of feature importance. If $v[i] \neq 0$, then the $i$th component of the representation $h_k(\cdot; W)[i]$ is important for making a prediction of the concept label. In other words, the bigger the absolute value of $v[i]$ the more information the corresponding neuron contains about the concept. In the ideal situation where $h_k(\cdot; W)$ does not encode concept information at all, we expect that $v = 0$. We propose to penalize the norm of the CAV vector in order to encourage less concept information, i.e., we introduce the following *adversarial CAV penalty* to the objective:

$$\mathsf{adv}_{C,k,\lambda}(W) = \|v_{C,k,\lambda}^*(W)\|^2. \qquad (3.2)$$

We emphasize that this choice is purely heuristic, intuitively we expect it to "push" the concept activation vector towards the origin.

---

[1] We remark that Arjovsky & Bottou (2017) has a whole section dedicated to modified loss functions as ways of mitigating the problem with vanishing and unstable gradients.

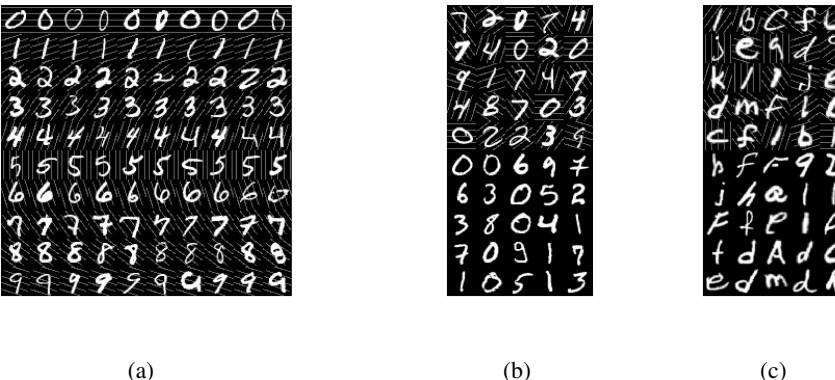

(a)            (b)            (c)

Figure 1: (a) Training dataset based on MNIST digits, where the angle of injected stripes is determined by the label: for a digit $j \in \{0, ..., 9\}$, the striped pattern is rotated by $\pi j/5$ radians (b) Concept dataset based on MNIST digits: the concept examples contain stripes (upper half) and the outer examples do not (bottom half); (c) Concept dataset based on EMNIST letters, where we introduce stripes at random angles, with $j$ in $\pi j/5$ uniformly drawn from $0, \ldots, 9$.

**Mini-batch optimization.** For stochastic gradient descent, we evaluate the terms in the objective

$$L_{DS}(W, \theta) + \gamma \mathsf{adv}_{C,k,\lambda}(W)$$

by averaging over a mini-batch. Here, $L_{DS}(W, \theta)$ is the downstream task loss, for instance, as in the first term of (2.2) ($\frac{1}{N}\sum_{i=1}^{N} \ell(f(h(X_i)), Y)$). For the adversarial penalty, we replace the sample averages in Eq. (A.1) and (A.2) with batch averages. However, we require a larger batch size when evaluating $\mathcal{D}_{0,\lambda}$, since it needs to be inverted in Eq. (A.1). We also notice that the inversion of a large matrix is not computationally tractable in practice. For example, the output of some intermediate layers of ResNet50 can reach up to 800K dimensions. In this case, even storing the square matrix $\mathcal{D}_{0,\lambda}$ in the memory is not possible. Using a batch average significantly speeds up the computation, and we propose a simplified procedure that does not require computing $\mathcal{D}_{0,\lambda}$ directly and replaces the matrix inverse with a linear solve operation, see Section B.

**Batch normalization.** Notice that the concept activation vector $v^*_{C,k,\lambda}$ can be shrunk by simple scaling of the representation output $h_k \rightarrow \alpha h_k$ (for instance, if the last layer is a convolution with ReLU activation). One way to avoid this is to equip $h_k(\cdot; W)$ with a batch normalization layer at the top. In the experiments, we make sure that this is the case.

## 4   DEEP CONCEPT REMOVAL

By default, the penultimate layer of the neural network is considered as the representation. However, in Section 4.2.2 of Kim et al. (2018), they demonstrate that a concept can be encoded in deeper layers as well. Motivated by this observation, we propose to apply adversarial CAVs simultaneously to a set of deep layers, rather than only to a penultimate one. That is, we fix some number of layers $k_1, \ldots, k_m$ and the hyperparameters $\lambda_1, \ldots, \lambda_m$, and we add up the total adversarial penalty

$$\mathsf{adv}_{C,k_1,\lambda_1}(W) + \cdots + \mathsf{adv}_{C,k_m,\lambda_m}(W).$$

We call this approach *Deep Concept Removal*. This however leaves us with a difficult choice of which exactly layers we should choose, since there can be many when we use a deep net. We are hoping to find an *interpretable* way of choosing these layers, without the need to select them through expensive search.

To find an answer, we take experimental approach, by conducting a simple case study. Specifically, we want to answer the following research questions:

- RQ1. *Does applying adversarial CAVs to not only the penultimate layer but also deeper and wider layers lead to more effective concept removal? What should be the choice?*

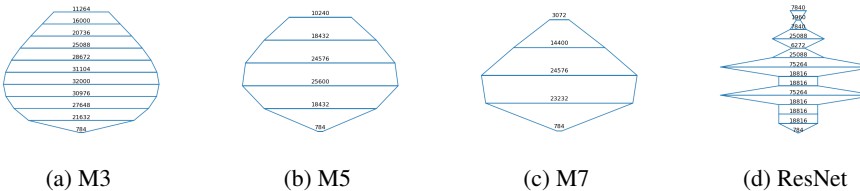

| (a) M3 | (b) M5 | (c) M7 | (d) ResNet |

Figure 2: Comparison of layer widths for CNN models M3, M5, and M7. Each horizontal line in the chart represents the width of a layer in the respective model, with the bottom layer corresponding to the MNIST input dimension of 784 (28 x 28). Note that the final fully connected (FC) layer is not depicted in the figures.

- RQ2. *Can the concept dataset be defined on out-of-distribution images, while maintaining the effectiveness of the deep concept removal method?*

Following a rolling example of "stripes" concept in Kim et al. (2018), we generate a modified MNIST dataset by injecting a striped pattern into the background of each digit, with the angle of the stripes dependent on the class label, Figure 1a. This dataset allows us to test the effectiveness of our deep concept removal methods by evaluating the models on the original MNIST test split. We first consider the most straightforward way to define the concept dataset — by using the same MNIST digits with and without stripes, see Figure 1b[2].

For experimenting, we consider a variety of convolutional neural networks (CNNs) of different shapes, see Figure 2. The first three networks 2a, 2b, 2c are simple feedforward CNNs which we borrow from An et al. (2020). Network 2d is a resnet-type network with 14 layers He et al. (2016). For each of them, we show the dimension of output of each layer, with the the width of each of them proportional to that dimension.

**To address RQ1,** we apply adversarial CAVs to different combinations of layers within these four networks. We first conduct an experiment, where we only use adversarial CAV at the penultimate layer with networks 2a, 2b, 2c. While the test performance for 2a and 2b can go up to 98%, it drops down to 84% for network 2c. What makes the latter different from the former two? We observe that the contraction of intermediate representations is more pronounced in model 2c, in the sense that the dimensionality of the output of the middle layers is much larger than that of the penultimate layer. We show that we can fix the performance of network (c) by including the widest layer that goes in the middle, which makes it also around 98%. See Figures 4a, 4b.

We note that these results partially conform with the observations made by Bau et al. (2017). They argue that wider networks are encouraging more disentangled concept. We can speculate that contraction in the middle layers results in entanglement of the concept with other features, so that when we remove the concept information, we can as well remove useful information and harm the performance. Based on our observations and observations of Bau et al. (2017), we propose to use the following rule to determine which layers we need to include.

**Rule.** *Apply the adversarial CAVs to (most) layers that precede contraction, which are the layers whose output has total dimension significantly larger than that of the consequent layer.*

We extend our analysis to a residual-type network (ResNet, He et al. (2016)), known for its structure of bottleneck blocks that intertwine contracting and expanding layers. This network has 6 layers that precede contraction: 4, 7, 8, 10, 11, 13 (we include the penultimate layer 13 since it precedes the contracting softmax layer). To further confirm our rule, we consider a large number of subsets layers and train such ResNet with Deep Concept Removal. The results are reported below in Table 1. Although the test error varies significantly, we notice that a good indication of a successful concept removal is that the training error is comparable with the test, that is, we generalize to out-of-distribution images without the stripes. And although there are some "outliers" to our statement,

---

[2]Such set-up is equivalent to domain adaptation, where we have labeled instances from the training domain (in our case, striped digits) and unlabeled images from the test domain (the original MNIST digits) Ganin & Lempitsky (2015).

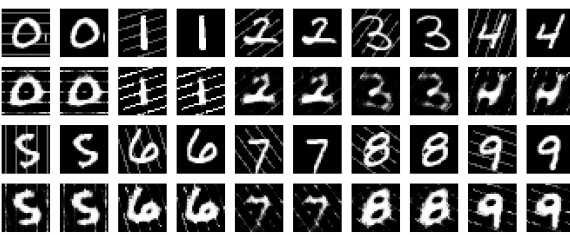

Figure 3: Output of the decoder trained with Concept Removal. The 1st and 3rd rows (from top) show examples of input with and without stripes. The image directly below each of them is the output of the decoder.

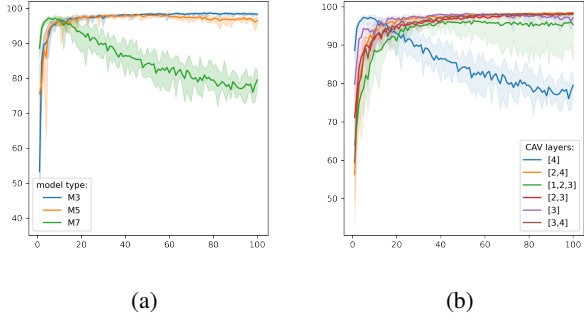

(a)                                           (b)

Figure 4: (a) Performance of different models on striped MNIST with adversarial CAV applied to the penultimate layer. (b) Performance of model M7 on striped MNIST with adversarial CAV applied to different sets of layers.

Figure 5: Comparison of test accuracy of concept removal with concept dataset based on MNIST and EMNIST. The x-axis is the number of epochs.

we can confidently say that including at least four layers from the list 4, 7, 8, 10, 11, 13 guarantees that we will remove the concept successfully and achieve an acceptable performance on test.

| Layers (High) | test | train | Layers (Medium) | test | train | Layers (Low) | test | train |
|---|---|---|---|---|---|---|---|---|
| 1, 7, 8, 11, 13 | 95.6 | 94.4 | 2, 8, 10, 11 | 79.8 | 98.9 | 1, 2, 9, 12 | 47.0 | 100.0 |
| 4, 7, 8, 10, 11, 13 | 93.5 | 90.9 | 5, 6, 7, 13 | 74.5 | 97.4 | 5, 6, 9, 12 | 35.5 | 100.0 |
| 4, 7, 10, 13 | 90.6 | 90.9 | 5, 7, 8, 9, 11 | 57.7 | 100.0 | 1, 6, 11 | 34.8 | 100.0 |
| 4, 7, 8, 10, 13 | 89.4 | 93.2 | 5, 7, 10 | 50.9 | 100.0 | 2, 4, 12 | 32.6 | 100.0 |
| 4, 7, 10, 11, 13 | 86.1 | 90.5 | 3, 7, 8 | 49.8 | 100.0 | 3, 5, 9 | 19.7 | 100.0 |

Table 1: MNIST experiment using a small ResNet. Each experiment corresponds to our adversarial CAV applied to a different set of layers. Layers preceding contraction are highlighted in cyan, including the penultimate layer. We report train and test accuracies after training for 100 epochs, averaged over three seeds.

**To address RQ2,** we additionally consider a concept dataset for the "stripes" concept based on EMNIST letters, as shown in Figure 1c. For instance, we look at a case where the concept is defined through instances that are not directly related to the original downstream task, which is closer to set-up in Kim et al. (2018). We compare the performance for the two concept datasets described above. The test performance is presented in Figure 5. Although the performance drops for EMNIST dataset, it remains adequate: we can see that the performance on the EMNIST concept drops by only around 3% with a slightly increased fluctuation of the last epoch. In addition, we can use a trained decoder to visually inspect removal of the stripes, see Figure 3. It is important to note that, with this concept dataset, the training algorithm never encounters the original MNIST digits without stripes,

whether labeled or unlabeled. To the best of our understanding, this is a novel problem setup that has not been previously explored in invariant representation learning literature.

We postpone all technical details of the experiments described above to Section C.1 in the appendix.

## 5 ROBUSTNESS TO DISTRIBUTIONAL SHIFTS AND OUT-OF-DISTRIBUTION GENERALIZATION

Distributionally robust optimization (DRO) concerns with classification problems, where the correlation between target variable $Y$ and an attribute $A$ changes during test. We focus on optimizing the *worst group accuracy*, which corresponds to the worst-case correlation shift in test Sagawa et al. (2019). Here, a group is a sub-population of data, which corresponds to a particular realization $(Y = y, A = a)$. The worst group accuracy of a classifier $\hat{Y}(X)$ reads as follows

$$\min_{(a,y)} \Pr\left(\hat{Y}(X) = Y | A = a, Y = y\right).$$

Usually, DRO tackles the situation where variables $Y$ and $A$ are spuriously correlated. When this occurs, some subgroups have low observation probabilities, making generalization difficult. An additional challenge is that the attributes are often not annotated, making it difficult to optimize the worst-group error.

Our concept removal approach can be used in cases where $A$ is binary, i.e. the attribute is an indicator of concept class membership. By using our approach, we hope to learn a representation that is transferable between subgroups $(A = 0, Y = y)$ and $(A = 1, Y = y)$. This can improve performance on the subgroup that has fewer observations. In addition, our approach does not require attribute annotation, rather it relies on an additional concept dataset to infer the concept directions.

We use two popular DRO benchmark datasets for evaluation:

- **Celeb-A** Liu et al. (2015) features aligned celebrity faces with annotated attributes (e.g. hair color, gender, wearing makeup, glasses, etc.) Our task is to classify blond hair. The spurious attribute is gender (male or not male). For the concept dataset, we select only celebrities with brown hair.

- **Waterbirds** Sagawa et al. (2019) contains bird images on different backgrounds. The task is to classify the birds as waterbirds or landbirds. The spurious attribute is the background (water or land). The benchmark includes training, validation, and test splits. The validation split is used for the concept dataset, allowing us to "decorrelate" the bird type and background.

- **CMNIST** Arjovsky et al. (2019) is an MNIST variant incorporating a spurious color feature. Labels are binarized (0 for digits 0-4, 1 for digits 5-9), with 25% label noise introduced via random flipping. Digits are colored red or green, with the color 90% correlated with the label, making color a stronger predictor than digit shape. For concept dataset, we used EMNIST letters with randomly added colors.

- **Striped MNIST** is our own adaptation of the original MNIST considered in Section 4, see Figure 1a. For the DRO experiment, we add the label-dependent stripes with probability 95%. This modification creates a dataset with 20 distinct groups, each corresponding to a combination of the 10 classes and the presence or absence of stripes. For concept dataset, we used EMNIST letters with and without stripes, Figure 1c.

We compare our method with the following popular baselines:

- **Empirical Risk Minimization (ERM)** optimizes the average error.

- **Just Train Twice (JTT)** Liu et al. (2021) is a two-phase method for handling spurious factors without annotation. It trains the model with ERM in the first phase to recognize the "short-cut" feature $A$; in the second phase, the model is fine-tuned by upweighting the error set to improve the accuracy of the smaller groups, all without requiring attribute annotations.

- **Group-DRO (gDRO)** is a method introduced by Sagawa et al. (2019). It requires group annotations and is designed to optimize the worst-group error directly. It is typically used as an upper bound for methods that do not use this information.

Similar to JTT, our method does not require attribute annotations during training. However, we additionally require a concept dataset to specify what information needs to be removed. In all cases, we use validation data with labeled groups for model selection. Table 2 compares our method's

Table 2: Test accuracies of different methods averaged over 5 seeds. Results for JTT are from Liu et al. (2021) and for ERM and gDRO from Idrissi et al. (2022). The group column indicates whether the method uses group attributes. Unlike JTT and ERM, our method uses additional information in the form of concept dataset.

| Method | Groups | Worst-group accuracy | | | |
|--------|--------|--------|-----------|--------|-------------|
| | | Celeb-A | Waterbirds | CMNIST | StripedMNIST |
| ERM | no | 79.7(3.7) | 85.5(1.0) | 0.2(0.3) | 93.0(1.2) |
| JTT | no | 81.1 | **86.0** | **71.5(0.7)** | 91.1(1.5) |
| Ours | no* | **83.9(0.7)** | 68.2(2.6) | 57.4 (16.8) | **96.5(0.8)** |
| gDRO | yes | **86.9(1.1)** | **87.1(3.4)** | **72.7(0.5)** | 96.0(0.9) |

performance on test with the baseline methods. We include training and model selection details in the Appendix, Section C.2.

The experiments demonstrate that our concept removal approach achieves comparable results with state-of-the-art DRO methods that do not use group labels on the Celeb-A dataset. However, on the Waterbirds dataset, our approach does not achieve competitive accuracy. Our results suggest that our concept removal approach is more effective at removing higher-level features, while lower-level features are deeply ingrained in the representations and harder to remove. This could explain why our method struggles to achieve comparable accuracy on the Waterbirds dataset. A similar comparison can be made between CMNIST and Striped MNIST, where stripes represent a more high-level concept.

## 5.1 Out-of-distribution generalization

Our experiments further show that removing concepts leads to features that are transferable not just to small sub-populations, but even to out-of-distribution data. To demonstrate this, we make a modification to the Celeb-A dataset by removing all Blond Males from the training set. Due to the space constraints, we postpone these results to Appendix, Section C.3.

## 6 Fair representation learning

In fair representation learning, one wants to train a representation that is oblivious of a sensitive attribute. Whether it is oblivious is typically measured through statistical dependence, due to its rigorous formulation Madras et al. (2018). However, with our method, we can train a classifier that is oblivious of the sensitive attribute in terms of interpretability method of Kim et al. (2018).

To demonstrate this, let us consider the problem of classifying professors against primary school teachers, Figure 6. We evaluate the importance of the gender concept (sensitive), as well as two other concepts that may be useful for distinguishing professors and primary school teachers, namely, eyeglasses and young. Using TCAV Kim et al. (2018), we compare the results of a model trained with and without concept removal, i.e. the standard ERM. In the Appendix, Section C, we provide a detailed description of how we selected examples for each of the concepts thanks to the annotations "Male", "Eyeglasses", "Young" in the Celeb-A dataset. We also describe in detail how we obtain the images of professors and teachers, and how we carry out the training process. Additionally, we show how the importance of these three concepts changes with the removal of the gender concept.

We compare our method as well as with standard Empirical Risk Minimization (ERM). The results are presented in Figure 7, where the leftmost graph displays the TCAV score for all three concepts, including also the scores for the ERM (represented by a dashed line). The TCAV score indicates that all three concepts are close to being 100% important (a value of 0 for the young concept corresponds to negative importance). It is worth noting that the TCAV score does not measure the extent to which a concept is important. In the middle plot of Figure 7, we show the sensitivity score (Eq. 2.1), which decreases with the application of concept removal for each of the three concepts. However, the score of the concepts "Young" and "Eyeglasses" is significantly higher than that of gender. On the other hand, for ERM, gender is the most important concept according to this score. The rightmost graph also shows the accuracy of the concept classifier for each of the three concepts. It can be observed

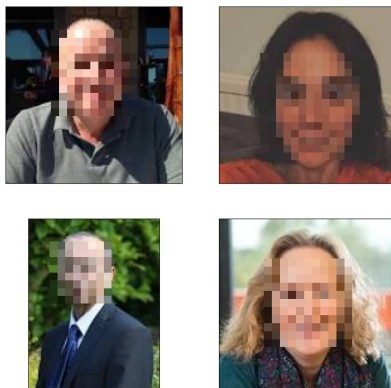

Figure 6: Examples of primary school teachers (top row) and professors (bottom row).

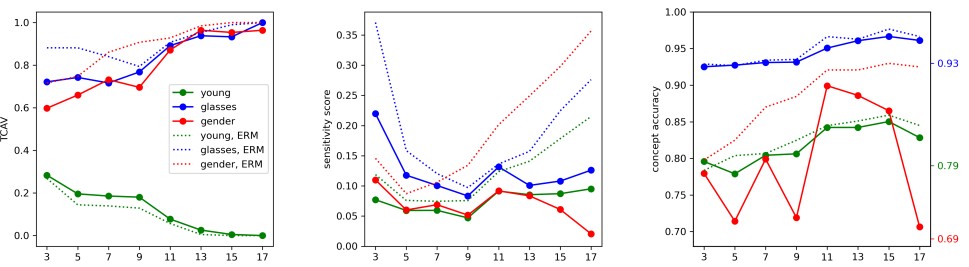

Figure 7: **Left** to **Right**: TCAV score calculated at different layers; the sensitivity score; accuracy of a linear estimator for each concept. Ticks on the right (rightmost graph) correspond to chance.

that for ERM, gender (red) can be classified with high accuracy, whereas for the concept removal model, the accuracy drops to the chance level when measured at the last layer. Additionally, we note that the accuracy for the other two concepts has decreased, but only slightly.

If we replace the concept dataset with the population data, which contains the same correlations between concept, as in the training, we end up learning fair representation in the traditional sense Madras et al. (2018). We show that this allows us to learn a fair representation (in a non-supervised manner), so that linear classifiers that fit on top (without any restrictions) satisfy group fairness measures. Notice that in this case, we do not require deep concept removal and it is sufficient to apply an adversarial classifier to the penultimate layer. We defer details to Section C.4 in the Appendix, where we conduct additional experiments with CelebA.

## 7    CONCLUSION AND LIMITATIONS

This work introduced a novel method to remove detrimental concepts when learning representation. We demonstrated that our method improves robustness to distributional shifts as well as out-of-distribution generalization. We also find applications in learning fair representations. Among limitations, we mention training time, which is a common issue in adversarial learning. Furthermore, although deep concept removal is not tied to a particular adversarial training algorithm, our current approach does not scale well with bigger models. We also leave the exploration of vision transformers and NLP applications for future research. We also emphasize that our method still requires validation when dealing with distributionally robust benchmarks. However, our approach requires a less aggressive model selection process, in particular, we always report the results for the model obtained after the last training epoch, unlike most of the existing DRO methods Idrissi et al. (2022).

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

# Appendix

## A DIFFERENTIATION OF CONCEPT ACTIVATION VECTOR

Here we derive the derivatives of $v^*_{C,k,\lambda}(W)$ defined in (3.1). We show that for any fixed $W_0$ and $v_0 = v^*_{C,k,\lambda}(W_0)$,

$$\frac{\partial v^*_{C,k,\lambda}}{\partial W^\top}(W_0) = -\mathcal{D}_{0,\lambda}^{-1}\left[\frac{1}{N_C}\sum_{i=1}^{N_C}\{\sigma_i - Y_i^C + \sigma_i(1-\sigma_i)h_k(X_i^C;W_0)v_0^\top\}\frac{\partial h_k(X_i^C;W_0)}{\partial W^\top}\right],$$
$$\tag{A.1}$$

where

$$\mathcal{D}_{0,\lambda} = \lambda I + \frac{1}{N_C}\sum_{i=1}^{N_C}\sigma_i(1-\sigma_i)h_k(X_i^C;W_0)h_k(X_i^C;W_0)^\top, \qquad \sigma_i = \sigma(v_0^\top h(X_i^C;W_0)). \tag{A.2}$$

Here, the role of quadratic regularization $\lambda\|v\|^2/2$ is twofold. On the one hand, it helps to compensate for the potentially small size of the concept dataset compared to the representation size. On the other hand, it improves the smoothness of $v^*_{C,k,\lambda}(W)$ as a function of $W$, specifically, we avoid exploding values when dealing with the inversion $\mathcal{D}_{0,\lambda}^{-1}$. We also notice that for $\lambda > 0$, the loss function is strongly convex and $v^*_{C,k,\lambda}(W)$ is uniquely defined.

Suppose, we have a function $f(v,w)$ of two parameters and we define $v^*(w) = \arg\min_v f(v,w)$. Given that for each fixed $w$, $f(\cdot,w)$ is strongly convex, this is equivalent to implicit function $\frac{\partial}{\partial v^\top}f(v^*,w) = 0$. The implicit function theorem then states that

$$\frac{\partial v^*(w)}{\partial w^\top} = -\left[\frac{\partial^2 v}{\partial v\partial v^\top}f(v,w)\bigg|_{v=v^*(w)}\right]^{-1}\frac{\partial^2}{\partial v\partial w^\top}f(v,w)\bigg|_{v=v^*(w)}. \tag{A.3}$$

We want to apply this formula to (3.1). Consider first,

$$\hat{v}(h_1,\ldots,h_N) = \arg\min_v \frac{1}{N}\sum_{i=1}^{N}\ell_{BCE}(\sigma(h_i^\top v);Y_i^C) + \frac{\lambda}{2}\|v\|^2.$$

We want to first calculate $\partial\hat{v}/\partial h_i^\top$ and then substitute $h_i = h(X_i^C;W)$ and apply the chain rule. The BCE loss with sigmoid logits looks as follows,

$$\ell_{BCE}(\sigma(h^\top v);Y) = -Y\log\sigma(h^\top v) - (1-Y)\log\left(1 - \sigma(h^\top v)\right).$$

Recall the standard sigmoid derivative relation $\sigma(x)' = \sigma(x)(1 - \sigma(x))$. Then, $\{\log\sigma(x)\}' = 1 - \sigma(x)$ and $\{\log(1-\sigma(x))\}' = -\sigma(x)$. By substituting $Y = 0$ and $Y = 1$ into the BCE loss, we can see that

$$\ell_{BCE}(\sigma(x);Y)' = \sigma(x) - Y.$$

We then calculate the following derivatives

$$\frac{\partial}{\partial v}\ell_{BCE}(\sigma(v^\top h);Y) = (\sigma(v^\top h) - Y)h,$$

$$\frac{\partial^2}{\partial v\partial v^\top}\ell_{BCE}(\sigma(v^\top h);Y) = \frac{\partial}{\partial v^\top}(\sigma(v^\top h) - Y)h = \sigma(v^\top h)(1 - \sigma(v^\top h))hh^\top,$$

$$\frac{\partial^2}{\partial v\partial h^\top}\ell_{BCE}(\sigma(v^\top h);Y) = \frac{\partial}{\partial h^\top}(\sigma(v^\top h) - Y)h = (\sigma(v^\top h) - Y)I + \sigma(v^\top h)(1 - \sigma(v^\top h))hv^\top.$$

Now we can plug this into (A.3). First, we calculate

$$\mathcal{D}_\lambda = \frac{\partial^2}{\partial v\partial v^\top}\left[\frac{1}{N_C}\sum_{i=1}^{N_C}\ell_{BCE}(\sigma(h_i^\top v);Y_i^C) + \frac{\lambda}{2}\|v\|^2\right] = \frac{1}{N_C}\sum_{i=1}^{N_C}\sigma_i(1-\sigma_i)h_ih_i^\top + \lambda I,$$

where we set $\sigma_i = \sigma(v^\top h_i)$. Applying (A.3) we get that

$$\frac{\partial\hat{v}}{\partial h_i} = -\frac{1}{N_C}\mathcal{D}_\lambda^{-1}\left((\sigma_i - Y_i^C)I + \sigma_i(1-\sigma_i)h_i\hat{v}^\top\right).$$

Finally, applying the chain rule, we obtain that

$$\frac{\partial v_C^*}{\partial W^\top} = \sum_{i=1}^{N} \frac{\partial \hat{v}}{\partial h_i} \frac{\partial h_i}{\partial W^\top} = -\mathcal{D}_\lambda^{-1} \left[ \frac{1}{N_C} \sum_{i=1}^{N_C} \left\{ (\sigma_i - Y_i^C)I + \sigma_i(1-\sigma_i)h_i[v^*]^\top \right\} \frac{\partial h_i}{\partial W^\top} \right].$$

## B    DETAILS OF IMPLEMENTATION

Here we describe a mini-batch implementation for computing gradients of the adversarial penalty (3.2).

Firstly, we notice that the gradients of $\mathsf{adv}_{C,k,\lambda}(W)$ are easier to compute. In particular, instead of calculating the gradient itself, we would like to produce a function that matches the derivatives of this penalty. The reason for this is that in most modern deep learning frameworks, the gradients are calculated automatically, and ideally we want to avoid intervening on that process. We consider the following function,

$$\widetilde{\mathsf{adv}}_{C,k,\lambda}(W; W_0, v_0) = -2x_0^\top \left[ \frac{1}{N_C} \sum_{i=1}^{N_C} (\sigma(v_0^\top h(X_i^C; W)) - Y_i^C)h(X_i^C; W) \right], \qquad \text{(B.1)}$$

where $x_0 = \mathcal{D}_{\lambda,0}^{-1} v_0$, $v_0 = v_{C,k,\lambda}^*(W_0)$, and $\mathcal{D}_{\lambda,0} = \mathcal{D}_{\lambda,0}(W_0)$ is defined in Eq. (A.2). Then, the identity holds

$$\frac{\partial}{\partial W^\top} \widetilde{\mathsf{adv}}(W_0; W_0, v_0) = \frac{\partial}{\partial W^\top} \mathsf{adv}(W_0). \qquad \text{(B.2)}$$

We check this identity at the end of this section. This term is added to a total loss to compute the gradient update. Let $W_0$ be the value of the parameter before the gradient update. In order to construct the function (B.1), we need to approximate the vectors $v_0 \approx v^*(W_0)$ and and $x_0 \approx \mathcal{D}_{0,\lambda}^{-1} v^*(W_0)$, where $\mathcal{D}_{0,\lambda} = \mathcal{D}_{0,\lambda}(W_0)$ can be very big. For instance, for ResNet50, the total dimension of the output of some intermediate layers reaches 800K. Calculating a 800K×800K matrix $\mathcal{D}_{0,\lambda}$ alone is not feasible on most hardware, let alone inverting it. Instead, we avoid estimating the matrix directly by using the Woodbury formula.

For every such gradient update of $W$, we conduct the following three steps:

**Input:**    `batch_size` — size of each batch of the data, same size is used to sample from training dataset and concept dataset, `cav_steps` — number of steps to update $v_0$. Overall, we draw `batch_size`×`cav_steps` dataponts from concept dataset for calculating (B.1).

**Step 1.**    Update the relevant CAV approximation (i.e., assuming we had changed $W$ in the previous gradient update). We conduct `cav_steps` gradient updates of $v_0$ with the cross-entropy objective from (3.1). We use a separate batch of data of size `batch_size` for each step. We fix the step to 0.001 in the experiments.

**Step 2.**    In this step we approximate the matrix $\mathcal{D}_{0,\lambda}$ by collecting the (detached) representations from the previous step. Denote the matrix containing these representations as $H$, so that it has a total of $M = $ `cav_steps` $\times$ `batch_size` rows. Then, a sample approximation of $\mathcal{D}_{0,\lambda}$ yields

$$\mathcal{D}_{\lambda,0} \approx \lambda I + \frac{1}{M}H^\top \Sigma H, \qquad \Sigma = \mathrm{diag}\{\sigma_1(1-\sigma_1), \ldots, \sigma_M(1-\sigma_M)\}, \qquad \sigma_j = \sigma(H[j,:]v_0).$$

Instead, we apply the Woodbury formula,

$$x_0 \approx \left( \lambda I + \frac{1}{M}H^\top \Sigma H \right)^{-1} v_0 = \lambda^{-1}v_0 - \lambda^{-2}M^{-1}H^T(\Sigma^{-1} + \lambda^{-1}HH^\top)^{-1}Hv_0.$$

**Step 3.**    Finally, given approximations of $x_0$ and $v_0$, we construct the function from (B.1). We replace the mean in the square brackets with a sample mean using only the last batch from Step 1.

**Remark 1.** *Training neural networks often benefits from using smaller batch sizes Keskar et al. (2016), but employing them to compute* (B.1) *can result in unstable training dynamics. We believe this instability stems from an inadequate approximation of $\mathcal{D}_{0,\lambda}$, not the sample average substitution in the square brackets of* (B.1). *To address this, we can increase* `cav_steps` *at the cost of higher computation, or raise the penalty parameter $\lambda$, risking underfitting of the linear concept classifier.*

**Check of Eq.** (B.2). In what follows we drop the indices $C, k, \lambda$ for sake of shortness. We have,

$$\frac{\partial}{\partial W^\top}(\sigma(v_0^\top h(X_i^C; W)) - Y_i^C)h(X_i^C; W)$$

$$= h(X_i^C; W)\frac{\partial}{\partial W^\top}\sigma(v_0^\top h(X_i^C; W)) + (\sigma(v_0^\top h(X_i^C; W)) - Y_i^C)\frac{\partial h(X_i^C; W)}{\partial W^\top}$$

$$= \left[\sigma(v_0^\top h(X_i^C; W)) - Y_i^C + \sigma'(v_0^\top h(X_i^C; W))h(X_i^C; W)v_0^\top\right]\frac{\partial h(X_i^C; W)}{\partial W^\top}.$$

Evaluating this at $W = W_0$ and using the notation $\sigma_i = \sigma(v_0^\top h(X_i^C; W_0))$ from Eq. A.1, we get that

$$\frac{\partial}{\partial W^\top}\widetilde{\mathsf{adv}}(W; W_0, v_0) = -2x_0^\top\left[\frac{1}{N_C}\frac{\partial}{\partial W^\top}\sum_{i=1}^{N_C}(\sigma(v_0^\top h(X_i^C; W)) - Y_i^C)h(X_i^C; W)\right]$$

$$= -2v_0^\top \mathcal{D}_{\lambda,0}^{-1}\sum_{i=1}^{N_C}\left[\sigma_i - Y_i^C + \sigma_i(1-\sigma_i)h(X_i^C; W_0)v_0^\top\right]\frac{\partial h(X_i^C; W_0)}{\partial W^\top}$$

$$= 2v_0^\top\frac{\partial v^*(W_0)}{\partial W^\top},$$

which evaluated at $v_0 = v^*(W_0)$ is exactly $\partial\|v^*(W)\|^2/\partial W^\top$.

## C DETAILS OF EXPERIMENTS

### C.1 MNIST EXPERIMENTS, SECTION 4

We generate the striped MNIST by adding a striped pattern to the background of each digit in the train split, making the stripes a decisive feature. The angle of the stripes depends on the class label; specifically, for a digit $j \in \{0, ..., 9\}$, the striped pattern is rotated by $\pi j/5$ radians, as illustrated in Figure 1a. To evaluate stripe removal effectiveness, we test our approach on the original MNIST test split. For the EMNIST-based concept dataset, we introduce stripes at random angles, with $j$ in $\pi j/5$ uniformly drawn from $0, \ldots, 9$.

In all experiments in Section 4 we use Adam optimizer with learning rate 0.001. For CAV step we use SGD with learning rate 0.001 without momentum. Furthermore, in all experiments we include a reconstruction error (pixelwise MSE) scaled to 10.0, the decoder is build on top of the representation as three fully connected layers, with 64, 256, and 784 units, respectively, the latter is then resized to the input shape 28x28. For simplicity, we take $\gamma = 1$ (the scaling factor of adversarial penalty) in all MNIST experiments. We note that including the reconstruction error in the objective is known to improve the training stability in the adversarial setting Zhao et al. (2017).

**Feedforward CNNs.** To include a bigger variety of models, we additionally examine three CNN models labeled M3, M5, and M7 from An et al. (2020), each consisting of consecutive Conv-Relu-BatchNorm blocks with kernel sizes 3, 5, and 7, respectively, followed by a fully connected layer. Model M3 has 10 convolutional layers with numbers of channels 32, 48, 64, 80, 96, 112, 128, 144, 160, and 176, while Model M5 comprises 5 layers with 32, 64, 96, 128, 160 channels, and Model M7 consists of 4 layers with 48, 96, 144, 192 channels. The total output dimensions of each layer of these models are visualized in Figures 2a-2c. We first evaluate the performance of each model when the adversarial CAV is applied at the penultimate layer, immediately before the final fully connected layer. We conduct five experiments with random seeds, and present the average test accuracy for 100 epochs in Figure 4a. We summarize our observations below:

- for the bigger model M3 the accuracy on the test consistently reaches 98%, which is closer to the original i.i.d. problem ($\approx 99\%$, see An et al. (2020));

- model M5 exhibits a similar behavior, although with a slightly worse accuracy, around 97%;

- model M7 completely fails on test: early during training the accuracy reaches 96% but with less consistency and, more importantly, the accuracy degrades as we train further, which makes it impossible to train adequately without the validation dataset.

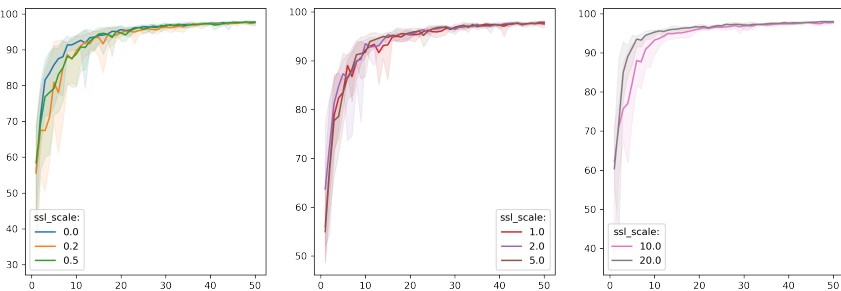

Figure 8: Performance of model M7 with adversarial CAVs at layers 2 and 4 and reconstruction scales 0.0, 0.2, 0.5, 1.0, 2.0, 5.0, 10.0, 20.0

What makes the model M7 different from models M3 and M5? We can see that the contraction of intermediate representations is more pronounced in M7. For the model M3, the output of the last convolutional layer is x1.6 times narrower than that of the layer before, and for model M5 it is x1.8 narrower. On the other hand, the last convolutional layer significantly lowers the dimension of the hidden representation from 14k to 3k which is more than three times.

In order to fix the problem with model M7, we propose to apply adversarial CAVs at multiple layers. The intuition is that we need to remove the concept from representation before it has the chance to entangle with other features. We try several subsets of layers to apply adversarial CAVs to and report the results in Figure 4b. We can see that including a deeper layer in addition to the last one (layer 4) fixes the problem. Interestingly, if we only include layer 3, the test accuracy can also be good and the performance does not degrade much with longer training. Another interesting observation is that the pair [2, 4] appears to work best in terms of stability and high test accuracy. We speculate that the reason that it works better than, for instance, the combination [3, 4] is that the 2nd convolutional layer has the highest output dimension, and the 3rd convolutional layer actually contracts it a little bit.

**ResNet and choice of layers.** Residual networks He et al. (2016) are simple and effective classifiers for relatively large images, typically, sized $224 \times 224$. Despite the fact that these models are not state of the art, they are still often used in practice, especially in the context of robustness Sagawa et al. (2019); Liu et al. (2021). Residual networks often have a lot of contracting and widening layers, namely, ResNet-50 and bigger. We consider a smaller network that has a similar to ResNet-50 shape, but only consists of 14 layers and 178K parameters, see Figure 2d where each layer dimension corresponds to the MNIST input size $28 \times 28$. Layers number 4, 7, 8, 10, 13 correspond to layers preceding contraction (we include the penultimate layer 13 since it precedes the contracting softmax layer). The results in Table 1 show adversarial CAVs applied to various combinations of layers. Although the test error varies significantly, we notice that a good indication of a successful concept removal is that the training error is comparable with the test, that is, we generalize to out-of-distribution images without the stripes. And although there are some "outliers" to our statement, we can confidently say that including at least four layers from the list 4, 7, 8, 10, 13 guarantees that we will remove the concept successfully and achieve an acceptable performance on test.

**Experiment with EMNIST-based concept dataset.** For the experiments we use model M7 with adversarial CAVs applied to layers 2 and 4. Other hyperparameters are same as in the above experiment.

**Importance of reconstruction error.** We run experiments with different scaling of the reconstruction error, see Figure 8. We take model M7 with adversarial CAV applied at layers 2, 4, both $\lambda = 0.1$. Notice how higher scaling encourages more stable learning during early epochs ($\leq 20$), while closer to the 100th epoch the reconstruction error becomes unnecessary. This can be useful for bigger pictures, where one typically starts with a pretrained model and uses a little bit of fine-tuning.

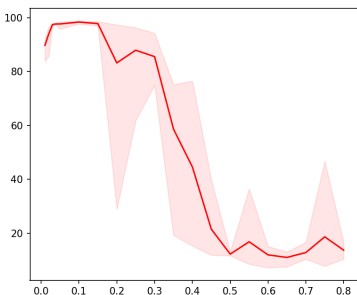

Figure 9: Accuracy on test vs. $\lambda$, with $\lambda$ ranging from 0.01 to 0.8. The line shows average over 5 seeds; the range fluctuations is depicted in semi-transparent color. We take model M3 with AdvCAV applied at the penultimate layer.

**Selection of $\lambda$.** In all of the above experiments, we chose CAV penalization $\lambda = 0.1$ in Eq.(3.1). We further conduct some experiments regarding the selection of $\lambda$, alternating it in the range $0.0..1.0$ and report the results in Figure 9.

### C.2 ROBUSTNESS TO DISTRIBUTIONAL SHIFTS, SECTION 5

We use ResNet50 and scale images to size 256, then center crop to size $224 \times 224$. Adversarial CAV is applied to 22th, 40th, and 49th convolutional layers. Each of these three layers is passed through avgpool. We choose $\lambda = 0.1$ for regularization. Additionally, we use adversarial CAV without any pooling applied to a flattened output of each of these layers with $\lambda = 5.0$. We choose weight decay in the range $\{0.001, 0.01, 0.1\}$. We use SGD with learning rate 0.0001 and momentum 0.9, and the learning rate decreases linearly from 0.01 to 0.0001 for the first 10 epochs. Celeb-A is trained for 100 epochs and Waterbirds for 500 epochs, with 5 seeds per configuration. Thanks to a more stable training, we always report the results for the *last epoch*. Hyperparameters are chosen based on the highest average validation error over the 5 seeds.

### C.3 OUT-OF-DISTRIBUTION GENERALIZATION

Our experiments further show that removing concepts leads to features that are transferable not just to small sub-populations, but even to out-of-distribution data. To demonstrate this, we make a modification to the Celeb-A dataset by removing all Blond Males from the training set. We treat blond males group as OOD. Similarly to the DRO expermients, we are only given examples of blond males in validation.

It is important to note that DRO methods that use labeled groups are not suitable for scenarios where one of the subgroups is not present. In table 3, we report both worst-group accuracy (among all groups except Blond Males) in test, and the OOD accuracy. We compare our method with JTT and ERM and report the results for both ResNet-18 and ResNet-50.

We train both ResNet-50 and ResNet-18 models for 50 epochs using SGD with learning 0.001 and momentum 0.9 and report the results for the last epoch. We run the experiment for weight decay in the range $\{0.001, 0.01, 0.1\}$ and number of CAV steps in the range 5, 10, 20. For ERM and JTT we pick the best epoch based on the validation error. Then, for each of the five seeds, we pick the best model based on validation error and report the average and std of the test. For our method, we only report the results after the last training epoch. For each configuration, we calculate the mean validation error over all seeds and choose the based one. We then report the average test error corresponding to this configuration.

For ResNet-50, adversarial CAV is applied to 22th, 40th, and 49th convolutional layers. Each of these three layers is passed through avgpool first. We choose $\lambda = 0.1$ for regularization. Additionally, we use adversarial CAV without any pooling applied to a flattened output of each of these

Table 3: Test accuracies of our method compared to ERM. Results averaged over 5 seeds, with standard deviation in the brackets. Worst-group ID shows the worst test accuracy among the groups Blond not Male, not Blond Male, not Blond not Male. OOD shows test accuracy for the group Blond Male.

| Method | Model | Worst-group ID | OOD |
|--------|-------|----------------|-----|
| ERM | ResNet50 | 80.8(2.4) | 76.8(6.0) |
| ERM | ResNet18 | 74.7(5.6) | 73.3(6.5) |
| JTT | ResNet50 | 81.3(4.8) | 73.5(4.1) |
| JTT | ResNet18 | 80.7(3.2) | 71.1(6.9) |
| Ours | ResNet50 | 77.9(4.5) | 80.3(4.0) |
| Ours | ResNet18 | 85.0 (3.4) | 81.1 (8.5) |

layers with $\lambda = 5.0$. For ResNet-18, we apply adversarial CAV to layers 5, 9, and 17 with the same parameters.

**Remark 2.** *We point out that unlike ERM and JTT, our method works better with the smaller model ResNet-18. We attribute this to the shape of ResNet-18, which contains smaller amount of contracting layers.*

## C.4 FAIR REPRESENTATION LEARNING

### C.4.1 DETAILS OF PROFESSORS AND PRIMARY TEACHERS EXPERIMENT.

**Data collection.** For this experiment we use face images of 1255 professors and 1357 primary school teachers in the UK, which we collect through Google image search. For primary school teachers, we search Google Images using queries that include "primary school teacher LinkedIn" followed by the name of a local borough or district. This information is sourced from the UK government website. For professors, we use the query "professor UNIVERSITY_NAME" and again, the list of university names is obtained from the government website. Next, we employ a pre-trained segmentation model (YOLOv4[3]) to eliminate images where the face does not occupy at least 40% of the frame. We then use another pre-trained model (DeepFace[4]) to filter out any images that cannot be recognized. DeepFace's gender classifier reveals that around 16% of professors are female and around 33% of teachers are male. It's important to note that DeepFace has been known to show bias towards males, so these numbers should be taken as rough estimates.

**Selection of concept datasets.** We construct each concept using examples from the Celeb-A dataset as outlined below:

1. Gender concept: images with "Male" attribute equal to 1 are used as examples of the concept class, while those with value 0 are outside of the concept class. To eliminate correlation between gender and hair color, only images with "Brown Hair" equal to 1 are used.

2. Young concept: to avoid correlation between "Eyeglasses" and "Young" within the "Male" subgroup, we sample 1/2 of the images from the group "not Male", 1/4 from the group "Male" & "Eyeglasses", and 1/4 from the group "Male" & "not Eyeglasses."

3. Eyeglasses concept: to avoid correlation between "Young" and "Eyeglasses" within the "Male" subgroup, we sample 1/2 of the images from the group "not Male", 1/4 from the group "Male" & "Young," and 1/4 from the group "Male" & "not Young."

**Training and model selection.** We train a ResNet-18 with SGD optimizer, batch size 256, weight decay 0.01, learning rate 0.001, and momentum 0.0. For concept removal, we apply adversarial CAVs to layers 5, 9, and 17 after passing them through AvgPool2d. In each case, we chose $\lambda = 0.1$ for CAV regularization in (3.1).

---

[3] https://github.com/Tianxiaomo/pytorch-YOLOv4
[4] https://github.com/serengil/deepface

In classification problems, one of the conventional ways to measure fairness is through group fairness. One example of group fairness is *demographic parity* (DP), which aims to equalize the probability of a positive outcome for a classifier in different groups corresponding to different realizations of the sensitive attribute. Given a classifier $\hat{Y}$, its demographic parity measurement reads as follows:

$$DP = \left| \mathbb{P}(\hat{Y} = 1 | A = 1) - \mathbb{P}(\hat{Y} = 1 | A = 0) \right|.$$

Another popular measure is *equalized odds* (EO), which takes into account the possible correlation between the sensitive attribute and the true target value $Y$. The equalized odds measurement is defined as follows,

$$EO = \max_{y \in \{0,1\}} \left| \mathbb{P}(\hat{Y} = 1 | Y = y, A = 1) - \mathbb{P}(\hat{Y} = 1 | Y = y, A = 0) \right|.$$

Both measures need to be close to zero for the algorithm to be considered "fair," with a smaller measure indicating a fairer algorithm. However, both measures are subject to the underlying distribution $\mathbb{P}$ of the data $(X, Y, A)$, which we assume to be the same distribution from which the training data is obtained.

In fair representation learning, we want to produce a representation of the data that can be used to construct fair classifiers. The need for a fair representation rather than a fair classifier can be two folds. Firstly, a representation can be used by a third party that is not trusted to produce fair classifiers. Secondly, representation learning can be done in an unsupervised manner, so that multiple fair classifiers can be fit later for multiple targets. In this case, the sensitive attribute is only required during the representation learning step, while the target labels are not required for representation learning step.

Madras et al. (2018) first emphasized the importance of fair representation learning without using target labels. They used adversarial training, similar to our method, as did subsequent research such as Adel et al. (2019); Feng et al. (2019); Xu et al. (2019). However, adversarial learning can be unstable for unsupervised fair representation learning, as noted in a recent survey Mehrabi et al. (2021). Recently, in the context of face recognition, Park et al. (2022) proposed an alternative method that uses alignment loss instead of adversarial learning.

We propose to use our Adversarial CAV as another alternative to existing adversarial techniques. If we assume that we only fit linear classifiers in the post-hoc step, we seek a representation $Z = Z(X)$ such that the demographic parity holds for any linear function $v^\top Z(X)$. To enforce this condition, we use our adversarial CAV and replace the concept dataset with the population distribution. That is, the pairs $(X, A)$ in the concept dataset must come from the same distribution, unlike in the case of the original problem of concept removal. We only apply the adversarial CAV at the last layer, which distinguishes our approach from the concept removal case. Below we demonstrate that our method achieves competitive performance, in particular, we compare with state-of-the-art method FSCL† (Park et al., 2022, Table 4) on Celeb-A dataset. We use "Male" attribute as sensitive.

Similar to Park et al. (2022), we learn the representation using SimCLR Chen et al. (2020a) with temperature 0.1 and MLP consisting of one hidden layer with 128 units and top layer with 128 units. We train for 1000 epochs with learning rate 0.0001 and momentum 0.9 using SGD. We add adversarial CAV penalty with $\lambda = 0.1$ only at the last layer passed through average pooling. We point out that for sake of faster training we only use 10K examples in the concept dataset (5K uniformly at random from each sensitive group). After the representation training stage, we fit a logistic regression classifier using the validation split and report the results on the test split for attributes "Attractive", "Big_Nose", "Bags_Under_Eyes". We report the accuracy along with the DP and EO metrics in Table 4. Although our method has lower accuracy, it is more conservative in terms of the fairness metrics in some cases.

# D    NOTE ON USING ADVERSARIAL LOSS WITH IMPLICIT GRADIENTS

In section 3 we claim that maximizing the loss of the adversarial problem cannot be improved with implicit gradients. We clarify this point here.

Table 4: Evaluation of fair representation learning. We report accuracy (Acc), EO, and DP for three different targets. The results are compared with another semi-supervised method FSCL† from Park et al. (2022).

| Method | Target: Attractive | | | Target: Bags_Under_Eyes | | | Target: Big_Nose | | |
|---|---|---|---|---|---|---|---|---|---|
| | Acc | DP | EO | Acc | DP | EO | Acc | DP | EO |
| ours | 70.9 (1.29) | 14.5 (3.7) | 7.3 (2.8) | 80.4 (0.4) | 3.2 (1.7) | 5.0 (1.8) | 80.0 (0.3) | 9.7 (2.8) | 7.6 (3.4) |
| FSCL† Park et al. (2022) | 74.6 (0.4) | not reported | 14.8(0.9) | not reported | not reported | not reported | 80.8 (0.2) | not reported | 6.1 (0.6) |

Suppose, following the standard adversarial learning approach, we use the adversarial penalty

$$\mathsf{adv}(W) = \max_v - \left[ \frac{1}{N_C} \sum_{i=1}^{N_C} \ell_{BCE}(\sigma(h_k(X_i^C; W)^\top v), Y_i^C) + \frac{\lambda}{2} \|v\|^2 \right]$$

instead of the square-norm penalty we define in (3.2). The maximum is attained for $v = v_{C,k,\lambda}^*(W)$.

In this case, the calculation of the gradient of $\mathsf{adv}(W)$ does not require gradients of $v_{C,k,\lambda}^*(W)$ thanks to the *envelope theorem*. In particular,

$$\frac{\partial}{\partial W} \mathsf{adv}(W) = \frac{\partial}{\partial W} \left( -\frac{1}{N_C} \sum_{i=1}^{N_C} \ell_{BCE}(\sigma(h_k(X_i^C; W)^\top v), Y_i^C) - \frac{\lambda}{2} \|v\|^2 \right) \Bigg|_{v=v_{C,k,\lambda}^*(W)} ,$$

where the function in the brackets depends on $v, W$, but we only have to take partial derivative w.r.t. to $W$. That is, the parameter $v$ can be considered as a constant.

