# OpenReview forum: "Deep concept removal"
_ICLR.cc/2024/Conference — Submitted to ICLR 2024_

### Official Review · Reviewer_EJVn · 2023-10-29

**Soundness:** 2 fair
**Presentation:** 3 good
**Contribution:** 2 fair
**Rating:** 5
**Confidence:** 3

**Summary:**

The paper achieves deep concept removal by penalizing the norm of concept activation vectors, and experimental results demonstrate the effectiveness of this method.

**Strengths:**

* The method in the paper is well-motivated.
* The paper is easy to follow.

**Weaknesses:**

* Lack of comparative methods. Many strategies for deep concept removal have already been proposed[1], such as adversarial concept removal mentioned by the authors. However, they did not compare their method with these classic approaches during the experimental phase.

* The performance of the method presented in the paper is also lacking. Table 2 shows a comparison between the authors' method and others, but it seems their method is not optimal, performing worse in some cases than methods without concept annotation.

* From the objective function of the method, it can be applicable to different downstream tasks. However, the paper only validated it on classification tasks. Is this method equally applicable to object detection or image generation tasks?

[1] Elazar, Yanai, and Yoav Goldberg. "Adversarial Removal of Demographic Attributes from Text Data." Proceedings of the 2018 Conference on Empirical Methods in Natural Language Processing. 2018.

**Questions:**

Is this method equally applicable to other downstream tasks, such as object detection or image generation tasks?

---

> ### Author Response · Authors · 2023-11-21
> **Reply to Reviewer EJVn**
>
> We thank you for taking the time to review our paper and kindly providing the comments. We hope we can bring some clarity with the answers below:
>
> > Lack of comparative methods. Many strategies for deep concept removal have already been proposed[1], such as adversarial concept removal mentioned by the authors. However, they did not compare their method with these classic approaches during the experimental phase.
>
> We want to emphasize that existing concept removal methods are all (to the best of our knowledge) post-hoc. For post-hoc methods, there is an issue stemming from the entanglement of the features in the last layer of representation, and removing the concept unavoidably removes correlated fatures. In fact, [1] theoretically proves that. We also note that existing methods are only designed to sensor the information, rather generalizing to non-concept instances. Our method is in-processing (and yes, more expensive), but it addresses a more difficult problem. For the sake of it, we implemented LEACE on Striped MNIST, results shown below. We use model M7 trained for 100 epochs with ERM, then apply LEACE, then train a logistic regression classifier.
>
> |                  | EMNIST concept dataset |                   | MNIST concept dataset |                   |
> |--------------------------|------------------------|-------------------|------------------------|-------------------|
> |                          | acc test               | acc train         | acc test               | acc train         |
> | LEACE                     | 0.59(0.01)             | 1 (0)             | 0.59(0.01)             | 1 (0)             |
>
> [1] Abhinav Kumar, Chenhao Tan, Amit Sharma. Probing Classifiers are Unreliable for Concept Removal and Detection
>
> > From the objective function of the method, it can be applicable to different downstream tasks. However, the paper only validated it on classification tasks. Is this method equally applicable to object detection or image generation tasks?
>
> Unfortunatelly, we do not have convincing experiments at the moment. We have tried applying the method to VAE, but there is no quantitative way to verify the results. Furthermore, it is not that trivial how to apply the method to the generative tasks. For instance, in the VAE case, the encoder itself can potentially add the concept back, even if the encoder successfully removed it. For instance, if we train on data that has some dependencies between the concept, it can use that information to infer whether the concept was in the input or not. In other words, both the decoder and encoder suffer from detrimental correlations, while our method is only suitable for removing concept from encoder.
>
> > The performance of the method presented in the paper is also lacking. Table 2 shows a comparison between the authors' method and others, but it seems their method is not optimal, performing worse in some cases than methods without concept annotation.
>
> We want to bring some clarity why we do the experiments in Table 2. Unlike in the controlled MNIST experiment, while concept removal assessment is not possible in dataset like Celeb-A.  The DRO problem hence can serve as a verification of concept removal. The other baselines in Table 2 address the problem of DRO directly. They are not concerned with concept removal.

---

### Official Review · Reviewer_pcxN · 2023-11-02

**Soundness:** 2 fair
**Presentation:** 2 fair
**Contribution:** 2 fair
**Rating:** 5
**Confidence:** 4

**Summary:**

This paper designs a novel concept removal method based on adversarial linear classifiers trained on a concept dataset, which aims to learn representations that do not encode certain specified concepts (e.g., gender etc.) Their proposed Deep Concept Removal further incorporates adversarial probing classifiers at various layers of the network, improving out-of-distribution generalization. Experiments on distributionally robust optimization (DRO) benchmarks demonstrates the advantage of their method.

**Strengths:**

+ originality: this paper proposed to utilize adversarial linear classifier to gain out-of-distribution robustness on the problem of concept removal

**Weaknesses:**

- lack of comparison with other concept removal baseline methods in Table 2

- lack of comprehensive results on Section 6, this paper does not show the advantage of their method on practical celebrity dataset.

- lack of ablation study on their training loss modules, for example, the effect of Penalty term of Eq. 3.2, and different values of \lambda of Eq. 3.1.

**Questions:**

- can the method be adapted to generative model where the concept removal task is of more importance?

---

> ### Author Response · Authors · 2023-11-17
> **Response to reviewer pcxN**
>
> We are thankful for the reviewer's comments, and we address each question below:
>
> >lack of comparison with other concept removal baseline methods in Table 2
>
> The problem with existing concept removal methods is they are only designed to sensor the information, rather generalizing to non-concept instances. For instance, the paper [1] theoretically proves that it cannot remove a concept without harming features that are correlated with it. For the sake of it, we implemented LEACE on Striped MNIST, results shown below. We use model M7 trained for 100 epochs with ERM, then apply LEACE, then train a logistic regression classifier:
>
> | EMNIST concept dataset|  |   MNIST concept dataset | |
> |---------------|----------------|----------------|--------------|
> |acc test        |  acc train      |    acc test        |  acc train      |
> |    0.59(0.01)|      1 (0)            |    0.59(0.01)   |  1 (0)|
>
> [1] Abhinav Kumar, Chenhao Tan, Amit Sharma. Probing Classifiers are Unreliable for Concept Removal and Detection
>
> > lack of comprehensive results on Section 6, this paper does not show the advantage of their method on practical celebrity dataset.
>
> We include Section 6 for sake of demonstration, in order to highlight the difference between fair representation and concept removal. In fair representation, we are concerned with statistical independence. In concept removal, we want to remove information associated with detrimental concepts, while keeping the useful information that correlates with it. While statistical dependence is subject to certain demographics (training set), a concept must be defined independently of the demographics.
>
> > lack of ablation study on their training loss modules, for example, the effect of Penalty term of Eq. 3.2, and different values of \lambda of Eq. 3.1.
>
> We include some ablation studies in Secion C.1, see end of section for choice of $\lambda$. We included Figure 9 incorrectly, fixed after revision. We apologize for this oversight.
>
> > can the method be adapted to generative model where the concept removal task is of more importance?
>
> We tried applying it to VAE. It is hard to assess the results it quantitatively. Furthermore, in VAE, even if the concept is removed by the encoder, it can be added by the decoder using correlated features.

---

### Official Review · Reviewer_fbvP · 2023-11-02

**Soundness:** 2 fair
**Presentation:** 2 fair
**Contribution:** 2 fair
**Rating:** 3
**Confidence:** 4

**Summary:**

This paper introduces a new method called Deep Concept Removal to remove undesirable concepts or features from learned representations in neural networks. The main idea of this method is to (a) use a concept dataset to learn a concept activation vector (CAV) in representation space and (b) penalize the norm of this CAV to down-weight the concept in representation space. The first set of experiments shows that this method can effectively remove a concept from synthetic MNIST data by (a) applying it to multiple wide layers and (b) using out-of-distribution concept datasets. The second set of experiments applies this method to (a) remove spurious concepts and improve worst-group accuracy in subpop robustness benchmarks and (b) reduces tcav sensitivity to spurious concepts in a fairness task.

**Strengths:**

- The background on concept activation vectors and adversarial concept removal is easy to follow for readers not familiar with the topics.
- The approach is simple and decouples model training and concept erasure, so one can erase concepts in a post-hoc manner using a concept dataset instead of requiring training datapoints to have concept labels.
- The experiments in S4 that study the connection between layers and concept removal effectiveness are insightful.

**Weaknesses:**

- The main experiment in S4 only demonstrates RQ1 and RQ2 on MNIST. The MNIST dataset is a good sanity check or starting point, but it is not enough to properly demonstrate the usefulness of this approach. A simple linear model can give 90% accuracy on the MNIST task and it is not representative of modern computer vision tasks. The experiments would be more convincing if the results hold on (a) “harder” tasks such as CIFAR-10 or CIFAR-100 and larger models (e.g., ResNet50 and ViTs)
- The experiments in S4 do not adequately support RQ2 (concept datasets can be OOD). The OOD concept dataset considered  (EMNIST) is in fact quite similar to the MNIST dataset. It would be useful to provide a more nuanced understanding of when OOD concept datasets fail to remove concepts for example.
- The experiments do not compare their results with relevant concept removal baselines such as LEACE (https://arxiv.org/abs/2306.03819) and kernel-space concept erasure (https://aclanthology.org/2022.emnlp-main.405/). There is no related work section either, so it’s hard for the readers to contextualize these findings.
- As mentioned by the authors, the regularization term (Eq 3.2) is purely heuristic, so it is unclear if this approach of downweighting the CAV will work in general.
- The deep concept removal method does not work on the Waterbirds dataset. In particular, it performs *significantly* worse than ERM. The authors do acknowledge this and hypothesize that their method may be more effective at removing high-level features than low-level features. However, this explanation is not convincing at all. What makes a concept high-level? Does this definition of high-level concept distinguish CelebA and “striped MNIST” concept from waterbirds? I am not sure if subpopulation robustness is the right task to evaluate concept removal. The images in the concept dataset with and without the concept may differ in many ways other than the presence of the concept, and this may lead to a spurious CAV.
- The figures are hard to parse (missing axes labels, legend too small). The writing in the second half (experiments, details, setup) can be significantly improved as well.

**Questions:**

- It would be interesting to know how sample efficient this method (as RQ3 in Section 4).

---

> ### Author Response · Authors · 2023-11-21
> **Response to Reviewer fbvP**
>
> We thank the reviewer for appreciating some parts of our paper and providing critical comments. Below we address these comments:
>
> > The main experiment in S4 only demonstrates RQ1 and RQ2 on MNIST. The MNIST dataset is a good sanity check or starting point, but it is not enough to properly demonstrate the usefulness of this approach. A simple linear model can give 90% accuracy on the MNIST task and it is not representative of modern computer vision tasks. The experiments would be more convincing if the results hold on (a) “harder” tasks such as CIFAR-10 or CIFAR-100 and larger models (e.g., ResNet50 and ViTs)
>
> This is a very good point. Unfortunately, we did not manage to make it work on problems where we are required to overfit. On all our MNIST experiments, in the cases where it works well, the train accuracy is usually smaller than test accuracy. Similarly, for the DRO we typically need to regularize in order to be robust. For both CIFAR-10 and CIFAR-100, we are required to overfit on train set in order to achieve competitive performance on test set.
>
> > The experiments in S4 do not adequately support RQ2 (concept datasets can be OOD). The OOD concept dataset considered (EMNIST) is in fact quite similar to the MNIST dataset. It would be useful to provide a more nuanced understanding of when OOD concept datasets fail to remove concepts for example.
>
> It is hard to say whether it is OOD enough, but there is not many good alternatives. We also want to emphasize that in this particular example, we require quite limited data. Learning invariant representations, to the best of our knowledge, requires access to the transformation [1, 2]. On the contrary, we only require some observations of the transformation on a relevant but not the same dataset. We also note that linear adversarial classifiers play a crucial role in this. If we use an MLP (equivalent to adversarial domain adaptation [3]), and no implicit gradients, the performance significantly drops. In both cases, whether the concept is based on EMNIST or MNIST. But there is also a bigger discrepancy between the two for the MLP:
>
> |         | EMNIST Concept Dataset |                | MNIST Concept Dataset |                |
> |---------|------------------------|----------------|-----------------------|----------------|
> |         | acc test               | acc train      | acc test              | acc train      |
> | MLP (DA)    | 18 (0.04)            | 1 (0)          | 0.44 (0.17)           | 1 (0)          |
>
> [1] Mitrovic et al. Representation learning via invariant causal mechanisims
>
> [2] von Kuegelgen et al. Self-supervised learning with augmentations provably isolates content from style
>
> [3] Ganin, Lempitsky. Unsupervised domain adaptation by backpropagation
>
> > The experiments do not compare their results with relevant concept removal baselines such as LEACE (https://arxiv.org/abs/2306.03819) and kernel-space concept erasure (https://aclanthology.org/2022.emnlp-main.405/). There is no related work section either, so it’s hard for the readers to contextualize these findings.
>
> It is known that such method suffer from feature entanglement and therefore cannot generalize between concept and non-concept examples [4]. In our quick implementation of LEACE we found that it does not work better than simple ERM for the StripedMNIST.
>
> [4] Abhinav Kumar, Chenhao Tan, Amit Sharma. Probing Classifiers are Unreliable for Concept Removal and Detection
>
> > The deep concept removal method does not work on the Waterbirds dataset. In particular, it performs significantly worse than ERM. The authors do acknowledge this and hypothesize that their method may be more effective at removing high-level features than low-level features. However, this explanation is not convincing at all. What makes a concept high-level? Does this definition of high-level concept distinguish CelebA and “striped MNIST” concept from waterbirds? I am not sure if subpopulation robustness is the right task to evaluate concept removal. The images in the concept dataset with and without the concept may differ in many ways other than the presence of the concept, and this may lead to a spurious CAV.
>
> This is a good question and we do not have a clear answer to it. However, let us give the following simple example. If we use a randomly initialized network, the penultimate layer most likely will contain information about the color, but not about the stripes (which can in fact be filtered out with random network [5]). We believe the same is true for Waterbirds, and often the algorithms for DRO utilize this information, by calling it a short cut feature. In that sense, it is very difficult to steer away the network from encoding that low-level information.
>
> [5] Ulyanov, Vedaldi, Lempitsky. Deep image prior.

---

### Official Review · Reviewer_QPRK · 2023-11-05

**Soundness:** 2 fair
**Presentation:** 2 fair
**Contribution:** 2 fair
**Rating:** 3
**Confidence:** 4

**Summary:**

This paper proposes a method to remove undesired concepts from neural network representations. This is done by an adversarial training process that alternates between training a concept classifier and a downstream task classifier. The novelty of their method lies in using an adversarial penalty based on TCAV, removing a concept of interest from multiple layers of the network. The paper tests its formulation across multiple settings.

**Strengths:**

1.	The proposed setting is novel and interesting. Concept removal is proposed as an intuitive extension to [1].
2.	The paper performs a comprehensive literature review and explains the relevant works needed to understand the paper in detail.
3.	The research questions proposed by the work are insightful for understanding how concepts are embedded in the network architecture. The finding that adversarial CAV helps most when applied to layers before contraction is important.
4.	The appendix provides extensive details about the implementation as well as the mathematical framework.

[1] Quantitative Testing with Concept Activation Vectors (TCAV)

**Weaknesses:**

1.	At a high-level, the paper seems to mainly combine two earlier works (Elazar et al, 2018 and Kim et al, TCAV, 2018), where it brings in the latter idea of a concept into the former’s framework. The novelty seems to be the choice of the adversarial loss, which is the norm of v* -- however, this is not well-motivated.
2.	The work seems to be more well-suited to bias removal than OOD generalization.
3.	Quantitative baselines that can convince the effectiveness of the proposed method are absent. In the example with StripedMNIST, test performance on images where stripes are at a different angle from those observed at training time are not presented.
4.	The results in Table 2 are not compelling enough. The disparity between the datasets where adversarial CAV performs poorly and where it performs better than the baselines is not satisfactorily addressed. “Our results suggest that our concept removal approach is more effective at removing higher-level features, while lower-level features are deeply ingrained in the representations and harder to remove” The distinction between lower-level and higher-level features seems arbitrary and unclear.
5.	Results from Fig 7 are not satisfactorily parsed. Why does the concept accuracy for “young” and “glasses” not deteriorate, whereas the concept accuracy for “gender” is reduced?
6.	OOD Generalization has been claimed, the numbers are only reported for Celeb-A, where the blond males have been called the ”unseen domain”. Firstly, most of these images are drawn from the same support as the original data, there is no covariate shift. Secondly, the so called ”domain invariant representation” has already been learned before hand by removing the concept or bias of gender. It is as if an oracle has given you information about the qualities of a domain-invariant generalization between both of the domains – this ideally should be deduced by the model, this makes the work weak.
7.	The limitation section fails to address some of the more pressing challenges identified in the work (for e.g. see #4)

My other concerns are presented in the questions section.

**Questions:**

1.	Is it trivial to extend the method to remove multiple concepts from the representations of a model? This would be closer to a realistic setting where the bias is generally a result of more than one concept.
2.	The adversarial penalty term has been used to remove unwanted concepts. Could a similar method be used to induce the usage of specific concept?
3.	[1] proposed concept activation regions to tackle the problem with TCAV, where the concept sets are not linearly separable despite the classes being linearly separable. How does this work address that issue?
4.	The formulation of the concept dataset following “...to avoid correlation between “Eyeglasses” and “Young” within the “Male” subgroup…” from appendix C.4.1 seems to imply a combination explosion as the number of concepts increases. Is this a limitation of the method?
5.	In Figure 3, for the images which do not have stripes as a concept, a reconstruction of those representations introduces stripes. This seems counterproductive.
6.	Please address the test accuracies being 100% in Table 1 even though the training accuracies are much lower.
7.	Please provide further details about the training of the decoder mentioned in RQ2. It is unclear what data is used along with the pixel wise MSE objective to train the decoder. Does the dataset include striped and unstriped images?
8.	In fig.7, are the various solid lines obtained from training multiple instances of the model where each instance has a specific concept removed?
9.	We request that in fig 4 (a), the epochs vs accuracy performance for ResNet MNIST be shown.

PS: Please provide axes labels to improve the readability of the graphs.

[1] Concept Activation Regions: A Generalized Framework For Concept-Based Explanations

---

> ### Author Response · Authors · 2023-11-17
> **Response to reviewer QPRK**
>
> We thank the reviewers for taking the time to read the paper and write valuable comments. We hope we can bring some more clarity with our responses below:
>
> > At a high-level, the paper seems to mainly combine two earlier works (Elazar et al, 2018 and Kim et al, TCAV, 2018)
>
>
> We do not agree with the characterization that we combine two methods (Elazar et al, 2018 and Kim et al, TCAV, 2018). Elazar et al (2018) and other adjacent work [1, 2] is concerned with post-hoc methods, ones that modify the representation produced by some method through a linear projection. Unfortunately, such methods are known to suffer from feature entangling. There is a paper dedicated to discussing this issue [3].
>
> Our method is in-processing, which means we do backpropagation into the models weights as compared to projection of the output of the penultimate layer in Elazar et al (2018).  We acknowledge that in-processing removal is much more expensive than a projection. However, there is simply no other method in literature that does the same. In the experiment with striped MNIST, we manage to learn invariant representation without access to transformation - concept dataset consist of EMNSIT letters. Furthermore, we believe that the characterization of which layers must be included is an interesting finding. Potentially, it suggests that in DNNs features get entangled thanks to contracting layers, which does conform with previous literature [4] (as we mention in p. 5).
> > Why does the concept accuracy for “young” and “glasses” not deteriorate, whereas the concept accuracy for “gender” is reduced?
>
> This example was handcrafted exactly to demonstrate the ability to remove one specific concept (gender), rather than everything that is correlated with it (glasses, young).
>
> [1] Tolga Bolukbasi, Kai-Wei Chang, James Y. Zou, Venkatesh Saligrama, and Adam T. Kalai. Man is to computer programmer as woman is to homemaker? De-biasing word embeddings.
>
> [2] Nora Belrose, David Schneider-Joseph, Shauli Ravfogel, Ryan Cotterell, Edward Raff, Stella Biderman. LEACE: Perfect linear concept erasure in closed form
>
> [3] Abhinav Kumar, Chenhao Tan, Amit Sharma. Probing Classifiers are Unreliable for Concept Removal and Detection
>
> [4] David Bau, Bolei Zhou, Aditya Khosla, Aude Oliva, and Antonio Torralba. Network dissection: Quantifying interpretability of deep visual representations.
>
> We also want to address some other questions:
>
> > Could a similar method be used to induce the usage of specific concept?
>
> That is a very interesting question. I believe it cannot. By removing one concept, we can only hope it will induce using other concepts, but there is no control over this.
>
> > OOD Generalization has been claimed, the numbers are only reported for Celeb-A, where the blond males have been called the ”unseen domain”. Firstly, most of these images are drawn from the same support as the original data, there is no covariate shift.
>
>
> In this experiment, the "blond males" subpopulation is completely excluded from the training set.
> > In Figure 3, for the images which do not have stripes as a concept, a reconstruction of those representations introduces stripes. This seems counterproductive.
>
> The goal of this Figure is to show that the reconstruction is the same for whether the stripes are present in the input or not. That is, we have learned invariant representation with respect to the presence of stripes. We do not claim that the decoder removes the stripes, but we claim that the encoder removes information of whether stripes are present or not.

---

### Meta-Review · Area_Chair_5EXc · 2023-12-06

**Metareview:**

This paper proposes a method to remove concepts/attributes from a model without compromising the performance. Overall, the paper is well-written and provides detailed descriptions of the approach. However, the reviewers found the novelty of the work to be limited and suggested that the approach should be viewed from the perspective of bias removal. More critically, multiple reviewers found that the comparisions with prior works in the experiment to be insufficient, abliet the authors' rebuttal. Additionally, some of the points asked by Reviewer QPRK were not addressed.

**Justification For Why Not Higher Score:**

None of the reviewers are willing to advocate for the paper.

**Justification For Why Not Lower Score:**

N/A

---

### Decision · Program_Chairs · 2024-01-16

Reject